# A Reduction-Based Framework for Conservative Bandits and Reinforcement Learning

**Yunchang Yang**[*]
Center for Data Science, Peking University
yangyc@pku.edu.cn

**Tianhao Wu**[*]
University of California, Berkeley
thw@berkeley.edu

**Han Zhong**[*]
Center for Data Sience, Peking University
hanzhong@stu.pku.edu.cn

**Evrard Garcelon, Matteo Pirotta, Alessandro Lazaric**
Facebook AI Research
{evrard, pirotta, lazaric}@fb.com

**Liwei Wang**
Key Laboratory of Machine Perception, MOE,
School of Artificial Intelligence, Peking University
International Center for Machine Learning Research,
Peking University
wanglw@cis.pku.edu.cn

**Simon S. Du**
University of Washington
ssdu@cs.washington.edu

## Abstract

We study bandits and reinforcement learning (RL) subject to a conservative constraint where the agent is asked to perform at least as well as a given baseline policy. This setting is particular relevant in real-world domains including digital marketing, healthcare, production, finance, etc. In this paper, we present a reduction-based framework for conservative bandits and RL, in which our core technique is to calculate the necessary and sufficient budget obtained from running the baseline policy. For lower bounds, we improve the existing lower bound for conservative multi-armed bandits and obtain new lower bounds for conservative linear bandits, tabular RL and low-rank MDP, through a black-box reduction that turns a certain lower bound in the nonconservative setting into a new lower bound in the conservative setting. For upper bounds, in multi-armed bandits, linear bandits and tabular RL, our new upper bounds tighten or match existing ones with significantly simpler analyses. We also obtain a new upper bound for conservative low-rank MDP.

## 1 Introduction

This paper studies online sequential decision making problems such as bandits and reinforcement learning (RL) subject to a conservative constraint. Specifically, the agent is given a reliable *baseline policy* that may not be optimal but still satisfactory. In conservative bandits and RL, the agent is asked to perform nearly as well (or better) as the baseline policy at all time. This setting is a natural formalization of many real-world problems such as digital marketing, healthcare, finance, etc. For example, a company may want to explore new strategies to maximize profit while simultaneously maintaining profit above a fixed baseline at any time, in order not to be bankrupted. See (Wu et al., 2016) for more discussions on the motivation of the conservative constraint.

Analogously to the non-conservative case, conservative bandit/RL problems also require us to balance exploration and exploitation carefully. Meanwhile, to ensure the obtained policies outperform the baseline policy, we need to provide a tractable approach to keep the exploration not too aggressive. Solving these two problems simultaneously is the key challenge in conservative bandits and RL.

Existing work proposed algorithms for different settings, including bandits (Wu et al., 2016; Kazerouni et al., 2016; Garcelon et al., 2020b; Katariya et al., 2019; Zhang et al., 2019; Du et al., 2020; Wang et al., 2021) and tabular RL (Garcelon et al., 2020a). However, lower bound exists only for the multi-

---

[*]equal contribution

armed bandit (MAB) setting (Wu et al., 2016), and there is no lower bound for other widely-adopted settings, such as linear bandits, tabular Markov Decision Process (MDP) and low-rank MDP. In Section 1.3, we provide a more detailed discussion of the related work.

For each of the different settings considered in the literature (i.e., multi-armed bandits, linear bandits, tabular MDPs), existing approaches rely on ad-hoc algorithm design and analysis of the trade-off between the setting-specific regret analysis and the conservative constraint. Furthermore, it is hard to argue about the optimality of the proposed algorithms because it would require clever constructions of the hard instances to prove the non-trivial regret lower bounds under the conservative constraint.

## 1.1 OUR CONTRIBUTIONS

In this paper, we address these limitations and make significant progress in studying the general problem of online sequential decision-making with conservative constraint. We propose a *unified framework* that is generally applicable to online sequential decision-making problems. The common theme underlying our framework is to calculate the necessary and sufficient budget required to enable non-conservative exploration. Such a budget is obtained by running the baseline policy (cf. Section 3). With the new framework, we obtain a novel upper bound on tabular MDPs, which improves the previous result. And we prove a new upper bound on low-rank MDPs. Also, we derive the first lower bounds for linear bandits, tabular and low-rank MDPs, which shows that our upper bound is tight.

**Lower Bounds.** For any specific problem (e.g., multi-armed bandits, linear bandits), our framework *immediately* turns a minimax lower bound of the non-conservative setting to a non-trivial lower bound for the conservative case (cf. Section 4). We list some examples to showcase the power of our framework for lower bounds. Full results are given in Table 1.

- We derive a novel lower bound for *multi-armed bandits* that works on a wider range of parameters than the one derived in (Wu et al., 2016). In particular, our lower bound shows a more refined dependence on the value of the baseline policy.
- We derive the *first* regret lower bound for conservative exploration in *linear bandits*, *tabular MDPs* and *low-rank MDPs*. These results allow to establish or disprove the optimality of the algorithms currently available in the literature.

We emphasize our technique for deriving lower bounds is simple and generic, so we believe it can be used to obtain lower bounds for other problems as well.

**Upper Bounds.** Our novel view of conservative exploration can also be used to derive high probability regret upper-bounds. When the suboptimality gap $\Delta_0$ and the expected return $\mu_0$ of the baseline policy are known, we show that the Budget-Exploration algorithm (Alg. 1) attains minimax optimal regret in a wide variety of sequential decision-making problems, when associated to any minimax optimal non-conservative algorithm specific to the problem at hand. In the more realistic (and challenging) scenario where $\Delta_0$ and $\mu_0$ are unknown, we show how to simply convert an entire class of algorithms with a sublinear non-conservative regret bound into a conservative algorithms with a sublinear regret bound. We obtain the following results, full details are given in Table 1.

- In the MAB setting, we obtain a regret upper-bound that matches our refined lower-bound, thus improving on existing analysis. In the linear bandit setting, we match existing bounds that are already minimax optimal.
- In the RL setting, we provide two novel results. First, we provide the first minimax optimal result for tabular MDPs, improving over (Garcelon et al., 2020a). Second, we derive the first upper bound for conservative exploration in low-rank MDPs. Our bound matches the rate of existing non-conservative algorithms though it is not minimax optimal. How to achieve minimax optimality in low rank MDPs is an open question even in non-conservative exploration.

Again, our reduction technique is simple and generic, and can be used to obtain new results in previously unstudied settings, like we did for low-rank MDPs.

## 1.2 MAIN DIFFICULTIES AND TECHNIQUE OVERVIEW

### 1.2.1 LOWER BOUNDS

The only lower bound for conservative exploration is by Wu et al. (2016) who followed a classical approach in the bandit literature. They constructed a class of hard environments and used an

| Setting | Lower Bound | Upper Bound |
|---|---|---|
| Multi-armed bandits | $\Omega\left(\sqrt{AT} + \frac{A\Delta_0}{\alpha\mu_0(\alpha\mu_0+\Delta_0)}\right)$ | $\widetilde{O}\left(\sqrt{AT} + \frac{A\Delta_0}{\alpha\mu_0(\alpha\mu_0+\Delta_0)}\right)$ |
| | $\Omega(\sqrt{AT} + \frac{A}{\alpha\mu_0})$
(Wu et al., 2016) [2] | $\widetilde{O}\left(\sqrt{AT} + \frac{A}{\alpha\mu_0}\right)$
(Wu et al., 2016) |
| Linear bandits | $\Omega\left(d\sqrt{T} + \frac{d^2\Delta_0}{\alpha\mu_0(\alpha\mu_0+\Delta_0)}\right)$ | $\widetilde{O}\left(d\sqrt{T} + \frac{d^2\Delta_0}{\alpha\mu_0(\alpha\mu_0+\Delta_0)}\right)$
This work and (Kazerouni et al., 2016; Garcelon et al., 2020b) |
| Tabular MDPs | $\Omega\left(\sqrt{H^3SAT} + \frac{SAH^3\Delta_0}{\alpha\mu_0(\alpha_0+\Delta_0)}\right)$ | $\widetilde{O}\left(\sqrt{H^3SAT} + \frac{SAH^3\Delta_0}{\alpha\mu_0(\alpha\mu_0+\Delta_0)}\right)$ |
| | | $\widetilde{O}\left(\sqrt{H^3SAT} + \frac{S^2AH^5\Delta_0}{\alpha\mu_0(\alpha\mu_0+\Delta_0)}\right)$
(Garcelon et al., 2020a) |
| Low Rank MDPs | $\Omega\left(\sqrt{d^2H^3T} + \frac{d^2H^3\Delta_0}{\alpha\mu_0(\alpha\mu_0+\Delta_0)}\right)$ | $\widetilde{O}\left(\sqrt{d^3H^4T} + \frac{d^3H^4\Delta_0}{\alpha\mu_0(\alpha\mu_0+\Delta_0)}\right)$ |

Table 1: Comparison of bounds for conservative decision-making. **Our contributions are reported in grey cells.** We denote by $T$ the number of rounds the agent plays (episodes in RL), $\alpha$ the conservative level, $\mu_0$ the expected return of the baseline policy[3], $\Delta_0$ the suboptimality gap of the baseline policy, $A$ the number of actions (or arms), $S$ the number of states and $d$ the feature dimension. The upper bounds hold both in the case $\Delta_0$ and $\mu_0$ are unknown since the lack of knowledge changes the regret only by a constant multiplicative factor (cf. Section 5).

information-theoretic argument to prove the lower bound. Construction of hard environments is highly non-trivial because one needs to incorporate the hardness from the conservative constraint. It is also non-trivial to generalize Wu et al. (2016)'s lower bound to other settings such as conservative linear bandits and RL because one will need new constructions of hard environments for different settings. We note that new constructions are needed even for non-conservative settings, because simply embedding the hard instances of MAB to other settings *cannot* give the tightest lower bounds. See, e.g., Chapter 24 of Lattimore & Szepesvári (2020) and Domingues et al. (2021).

In this paper, We use a completely different approach. Our key insights are 1) **relating the necessary budget to the regret lower bounds of non-conservative sequential decision-making problems**, and 2) obtaining sharp lower bounds in the conservative settings via **maximizing a quadratic function (cf. Equation** (6)). Comparing with the classical approach, our approach is simpler and more general: ours does not need problem-specific constructions and can automatically transform any lower bound in a non-conservative problem to the corresponding conservative problem. See Section 4 for details.

### 1.2.2 UPPER BOUNDS

**Improvement over Wu et al. (2016) when $\Delta_0$ is known.** When $\Delta_0$ is known, Wu et al. (2016) proposed an algorithm (BudgetFirst) which first plays the baseline policy for enough times and then plays an non-conservative MAB algorithm. However, their regret bound is not tight because their analysis on the required budget is loose: they accumulate enough budget to play $T$-**step** exploration where $T$ is the total number of rounds. Our main technical insight to obtain the tight regret bound is a sharp analysis on the required budget: by relating the minimax regret upper bounds of UCB algorithms, we show the required budget can be **independent of** $T$. See Section 5 and F for details.

**Sharp upper bounds with unknown $\Delta_0$.** When $\Delta_0$ is unknown, the paper by Wu et al. (2016), its follow-up papers (Kazerouni et al., 2016; Garcelon et al., 2020b; Zhang et al., 2019; Garcelon et al., 2020a), and our paper, all adopt the same algorithmic template: 1) build an online estimate on the lower bound performance of each possible exploration policy, and 2) based on the estimated lower bounds, choose an exploration policy or play the baseline policy.

---

[2]Although the lower bound in Wu et al. (2016) seems tighter, they require a condition $\frac{\Delta_0}{\alpha\mu_0+\Delta_0} \geq 0.9$. Under this condition, our lower bound is the same as theirs. Thus ours is more general. See Appendix E.

[3]In (Garcelon et al., 2020a), the upper bound scales with $r_b = \min_{s\in\mathcal{S},\rho_0(s)>0} V_1^{\pi_0}(s)$ (with $\rho_0$ the distribution of the starting state), the minimum of the baseline's value function at the first step over the potential starting states. Here, we assume there is a unique starting state hence $r_b = V^{\pi_0}$.

The key difference and the most non-trivial part in different papers is how to analyze $T_0$ (the number of times of executing the baseline policy). Existing works upper bound $T_0$ by relating it to the decision criterion for whether to choose the baseline policy or not. Since for different problem settings, the criteria have different forms, existing papers adopt different problem-specific analyses, and in some settings, the analyses are not tight (e.g., MAB and tabular RL). Our analysis approach is different from existing ones: we bound $T_0$ via **maximizing a quadratic function that depends on the minimax regret bounds of non-conservative algorithms and the conservative constraint**. See Section 5 for more details.

### 1.3 RELATED WORK

Non-conservative exploration has been widely studied in bandits, and minimax optimal algorithms have been provided for the settings considered in this paper (e.g. Lattimore & Szepesvári, 2020). The exploration problem has been widely studied also in RL but minimax optimal algorithms have not been provided for all the settings. For any finite-horizon time-inhomogeneous MDP with $S$ states, $A$ actions and horizon $H$, the minimax regret lower bound is $\Omega(\sqrt{H^3 SAT})$ (Domingues et al., 2021), where $T$ denotes the number of episodes. For any time-inhomogeneous low-rank MDP with $d$-dimensional linear representation, the lower-bound is $\Omega(\sqrt{d^2 H^3 T})$ (Zhou et al., 2020, Remark 5.8). While several minimax optimal algorithms have been provided for tabular MDPs (e.g. Azar et al., 2017; Zanette & Brunskill, 2019; Zhang et al., 2020a;b; Ménard et al., 2021), the gap between upper bound and lower bound is still open in low-rank MDPs, where LSVI-UCB (Jin et al., 2020) attains a $\widetilde{O}(\sqrt{d^3 H^4 T})$, while ELEANOR (Zanette et al., 2020) improves to $\widetilde{O}(\sqrt{d^2 H^4 T})$.

In conservative exploration, previous works focus on designing specific conservative algorithms for different settings. This conservative scenario was studied in multi-armed bandits (Wu et al., 2016), contextual linear bandits (Kazerouni et al., 2016; Garcelon et al., 2020b), contextual combinatorial bandits (Zhang et al., 2019) and tabular MDPs (Garcelon et al., 2020a). All these works focused on providing an upper-bound to the regret of a conservative algorithm. Other problems that have been considered in conservative exploration are combinatorial semi-bandit with exchangeable actions (Katariya et al., 2019) and contextual combinatorial cascading bandits (Wang et al., 2021). Du et al. (2020) have recently considered conservative exploration with sample-path constraint.

Our work is also related to safe bandits/RL (Amani et al., 2019; Pacchiano et al., 2021; Amani et al., 2021) and constrained RL (Altman, 1999; Efroni et al., 2020; Ding et al., 2020; 2021; Chen et al., 2020). The setting of safe bandits/RL is different from conservative bandits/RL. Specifically, the safety constraint requires that the expected cost at each stage is below a certain threshold. This constraint is stage-wise, and is independent of the history. On the contrary, the conservative constraint requires that the total reward is not too small. For the constrained MDP, the goal is to maximize the expected reward value subject to a constraint on the expected utility value (value function with respect to another reward function). In conservative RL, however, the agnet aims to maximize the expected reward value subject to the constaint that the (same) reward value is not significantly worse that of the baseline policy.

## 2 PRELIMINARIES

The objective of this section is to provide a unified view of the settings considered in this paper, i.e., multi-armed bandits, linear bandits, tabular Markov Decision Processes (MDPs) and low-rank MDPs. We use the RL formalism which encompasses the bandit settings.

**Notations.** We begin by introducing some basic notation. We use $\Delta(\cdot)$ to represent the set of all probability distributions on a set. For $n \in \mathbb{N}_+$, we denote $[n] = \{1, 2, \ldots, n\}$. We use $O(\cdot), \Theta(\cdot), \Omega(\cdot)$ to denote the big-O, big-Theta, big-Omega notations. We use $\widetilde{O}(\cdot)$ to hide logarithmic factors. We denote $A \gtrsim (\lesssim) B$ if there exists a positive constant $c$ such that $A \geq (\leq) cB$.

**Tabular MDPs.** A tabular finite-horizon time-inhomogeneous MDP can be represent as a tuple $M = (\mathcal{S}, \mathcal{A}, H, \{p_h\}_{h=1}^H, s_1, \{r_h\}_{h=1}^H)$, where $\mathcal{S}$ is the state space, $\mathcal{A}$ is the action space, $H$ is the length of each episode and $s_1$ is the initial state. At each stage $h$, every state-action pair $(s, a)$ is characterized by a reward distribution with mean $r_h(s, a)$ and support in $[0, r_{\max}]$, and a transition distribution $p_h(\cdot|s, a)$ over next states. We denote by $S = |\mathcal{S}|$ and $A = |\mathcal{A}|$. A (randomized) policy $\pi \in \Pi$ is a set of functions $\{\pi_h : \mathcal{S} \mapsto \Delta(\mathcal{A})\}_{h \in [H]}$. For each stage $h \in [H]$ and any state-action

pair $(s, a) \in \mathcal{S} \times \mathcal{A}$, the value functions of a policy $\pi$ are defined as:

$$Q_h^\pi(s, a) = \mathbb{E}\left[\sum_{h'=h}^{H} r_{h'}|s_h = s, a_h = a, \pi\right], \quad V_h^\pi(s) = \mathbb{E}\left[\sum_{h'=h}^{H} r_{h'}|s_h = s, \pi\right].$$

For each policy $\pi$, we define $V_{H+1}^\pi(s) = 0$ and $Q_{H+1}^\pi(s, a) = 0$ for all $s \in \mathcal{S}, a \in \mathcal{A}$. There exists an optimal policy $\pi^\star$ such that $Q_h^\star(s, a) = Q_h^{\pi^\star}(s, a) = \max_\pi Q_h^\pi(s, a)$ satisfy the optimal Bellman equations $Q_h^\star(s, a) = r_h(s, a) + \mathbb{E}_{s' \sim p_h(s,a)}[V_{h+1}^\star(s')]$ and $V_h^\star = \max_{a \in \mathcal{A}}\{Q_h^\star(s, a)\}$. Then the optimal policy is the greedy policy $\pi_h^\star(s) = \arg\max_{a \in \mathcal{A}}\{Q_h^\star(s, a)\}$.

**Low-Rank MDPs.** We assume that $\mathcal{S}, \mathcal{A}$ are measurable spaces with possibly infinite number of elements. For algorithmic tractability, we shall restrict the attention to $\mathcal{A}$ being a finite set with cardinality $A$. When the state space is large or uncountable, value functions cannot be represented in tabular form. A standard approach is to use a parametric representation. Here, we assume that transitions and rewards are linearly representable (Jin et al., 2020).

**Assumption 1** (Low-rank MDP). *An MDP $(\mathcal{S}, \mathcal{A}, H, p, r)$ is a linear MDP with a feature map $\phi : \mathcal{S} \times \mathcal{A} \to \mathbb{R}^d$, if for any $h \in [H]$, there exist $d$ unknown (signed) measures $\boldsymbol{\mu}_h = \left(\mu_h^{(1)}, \ldots, \mu_h^{(d)}\right)$ over $\mathcal{S}$ and an unknown vector $\boldsymbol{\theta}_h \in \mathbb{R}^d$, such that for any $(x, a) \in \mathcal{S} \times \mathcal{A}$, we have*

$$\mathbb{P}_h(\cdot \mid x, a) = \langle \boldsymbol{\phi}(x, a), \boldsymbol{\mu}_h(\cdot) \rangle, \quad r_h(x, a) = \langle \boldsymbol{\phi}(x, a), \boldsymbol{\theta}_h \rangle. \tag{1}$$

*Without loss of generality, we assume $\|\boldsymbol{\phi}(x, a)\| \leq 1$ for all $(x, a) \in \mathcal{S} \times \mathcal{A}$, and $\max\{\|\boldsymbol{\mu}_h(\mathcal{S})\|, \|\boldsymbol{\theta}_h\|\} \leq \sqrt{d}$ for all $h \in [H]$.*

Under certain technical conditions (e.g., Shreve & Bertsekas, 1978), all the properties of tabular MDPs extend to low-rank MDPs. In addition, the state-action value function of any policy $\pi$ is linearly representable in low-rank MDPs. Formally, for any policy $\pi$ and stage $h \in [H]$, there exists $\theta_h^\pi \in \mathbb{R}^d$ such that $Q_h^\pi(s, a) = \langle \phi(s, a), \theta_h^\pi \rangle$.

**Connection between RL and Bandits.** To have a unified view, we can represent a multi-armed bandit as a tabular MDP with $S = 1$, $A$ actions, $H = 1$ and self-loop transitions in $s_1$. In multi-armed bandits, we consider only deterministic policies so that $\Pi = \mathcal{A}$, then $V^\pi(s_1) = r(s_1, \pi(s_1))$ and the optimal policy is simply $\pi^\star = \arg\max_{a \in \mathcal{A}} r(s_1, a)$. Similarly, a linear bandit can be modeled through low-rank MDPs with $H = 1$. For generality, we allow the action space to be possibly uncounted and we define the value of a deterministic policy $\pi = a$ ($\Pi = \mathcal{A}$) as $V_1^\pi(s_1) = r_1(s_1, a) = \langle \phi(s_1, a), \theta_1 \rangle$. The optimal policy $\pi^\star$ is thus such that $\pi^\star = \arg\max_{a \in \mathcal{A}} \langle \phi(s_1, a), \theta_1 \rangle$. We refer the reader to Appendix A for details.

## 3 GENERAL FRAMEWORK FOR CONSERVATIVE EXPLORATION

With the unified view provided in the previous section, we can consider a generic sequential decision-making problem $\mathfrak{P}$ over $T \in \mathbb{N}^\star$ episodes. We consider the standard online interaction protocol where, at each episode $t \in [T]$, the learning agent $\mathfrak{A}$ selects a policy $\pi_t$, observes and stores a trajectory $(s_i, a_i, r_i)_{i \in [H]}$, updates the policy and restart with the next episode. We evaluate the performance of the learner through the *pseudo-regret*. Let $V^\pi = V_1^\pi(s_1)$ be the value function of a policy $\pi$, then the regret is defined as:

$$R_T(\mathfrak{P}, \mathfrak{A}) = \sum_{t=1}^{T} V^\star - V^{\pi_t}. \tag{2}$$

In conservative exploration, the learner aims to minimize the regret while guaranteeing that, at any episode $t$, their expected performance is (nearly) above the one of a baseline policy $\pi_0$. Formally, given a possibly randomized baseline policy $\pi_0 \in \Pi$ and a conservative level $\alpha \in [0, 1]$, the learner should satisfy w.h.p. that

$$\forall t \leq T, \quad \sum_{j=1}^{t} V^{\pi_j} \geq (1 - \alpha) t V^{\pi_0}. \tag{3}$$

We assume that the value of conservative policy $V^{\pi_0}$ is known to the agent. Such assumption can be seen in previous works such as Wu et al. (2016); Kazerouni et al. (2016); Garcelon et al. (2020b;a). This assumption is reasonable in practice because usually the baseline policy has been used for a long time and is well-characterized, and its value can be estimated using historical data. Even if we do not know the value of baseline policy, we can estimate it during the algorithm (e.g., Section 3.5 in Wu et al. (2016)), and we omit here for simplicity.

## 3.1 BUDGET OF A CONSERVATIVE ALGORITHM

Given the set of policies $\{\pi_t\}_{t \in [T]}$ selected by a conservative algorithm $\mathfrak{A}$, we can divide the episodes into the set $\mathcal{T}_0 = \{t \leq T \mid \pi_t = \pi_0\}$ and its complement $\mathcal{T}_0^c = \{t \leq T \mid \pi_t \neq \pi_0\} = [T] \setminus \mathcal{T}_0$. The set $\mathcal{T}_0^c$ denotes the episodes where the algorithm played an exploratory policy, i.e., it had enough budget to satisfy condition (3) through a policy $\pi_l \neq \pi_0$. This sequence of non-baseline policies $\{\pi_t\}_{t \in \mathcal{T}_0^c}$ defines a new algorithm $\widetilde{\mathfrak{A}}$, that we refer as the non-conservative algorithm. However, the algorithm $\mathfrak{A}$ is conservative therefore, for any $\delta > 0$ and $t \in [T]$, we have with probability at least $1 - \delta$ that $\sum_{l=1}^t V^{\pi_l} \geq (1 - \alpha)t V^{\pi_0}$. Hence, for any $t \in [T]$ we have:

$$\alpha V^{\pi_0} |\mathcal{T}_{0,t}| \geq \sum_{l \in \mathcal{T}_{0,t}^c} (1 - \alpha) V^{\pi_0} - V^{\pi_l}, \tag{4}$$

where $\mathcal{T}_{0,t} = \mathcal{T}_0 \cap [t]$ and $\mathcal{T}_{0,t}^c = \mathcal{T}_0^c \cap [t]$. Taking maximum over $t$ in Eq. (4), we have that with high probability the conservative algorithm $\mathfrak{A}$ is such that

$$\alpha V^{\pi_0} |\mathcal{T}_0| \geq \underbrace{\max_{t \leq T} \sum_{l \in \mathcal{T}_{0,t}^c} (1 - \alpha) V^{\pi_0} - V^{\pi_l}}_{= \mathcal{B}}.$$

The quantity on the right of the previous equation is exactly the amount of reward that the conservative algorithm $\mathfrak{A}$ has to collect by playing the baseline policy. Hence this quantity acts as a *conservative budget* $\mathcal{B}$. The higher it is, the more $\mathfrak{A}$ needs to play the baseline policy to satisfy the conservative condition. In other words, it is the least amount of reward that an algorithm needs to not violate the conservative constraint. We now extend this notion to any (non necessarily conservative) algorithm.

**Definition 1.** *For any $T \in \mathbb{N}^\star$, set of episodes $\mathcal{O} \subset [T]$ and arbitrary sequence of policies $\{\pi_t\}_{t \in \mathcal{O}}$, the budget of this sequence of policies is defined as:*

$$\mathcal{B}_T(\mathcal{O}, \{\pi_t\}_{t \in \mathcal{O}}) = \max_{t \in \mathcal{O}} \sum_{l \in \mathcal{O} \cap [t]} (1 - \alpha) V^{\pi_0} - V^{\pi_l}. \tag{5}$$

## 4 REGRET LOWER BOUND FOR CONSERVATIVE EXPLORATION

In this section, we leverage the framework introduced in Section 3 to build lower bounds for several problems. Our result is based on the notion of budget defined in Section 3. This notion is used to build an algorithm whose regret is a lower bound for any conservative algorithm.

**Theorem 1** (Conservative Exploration Regret Lower Bound)**.** *Let's consider a decision-making problem $\mathfrak{P}$ over $T$ steps, a conservative level $\alpha \in [0, 1]$, a baseline policy $\pi_0$, an algorithm $\mathfrak{A}$ and $\delta \in (0, 1)$. We assume that:*

- *Lower-bound for non-conservative exploration. There exists a $\xi \in \mathbb{R}_+$ and $T_0 \in \mathbb{N}$ such that for any algorithm $\mathfrak{A}'$ there exists an environment (instance of $\mathfrak{P}$) such that with probability at least $1 - \delta$, $R_T(\mathfrak{P}, \mathfrak{A}') \geq \xi\sqrt{T}$ for $T \geq T_0$.*
- *$\mathfrak{A}$ is conservative. The algorithm $\mathfrak{A}$ is conservative, that is to say with probability at least $1 - \delta$ for any $t \leq T$, $\sum_{l=1}^t V^{\pi_l} \geq (1 - \alpha)t V^{\pi_0}$.*

*Then, there exists an environment (instance of problem $\mathfrak{P}$) and $T_0 \in \mathbb{N}$ such that with probability at least $1 - \delta$ and $T \geq T_0$:*

$$R_T(\mathfrak{A}, \mathfrak{P}) \gtrsim \max\left\{\xi\sqrt{T}, \frac{\xi^2 \Delta_0}{\alpha V^{\pi_0}(\alpha V^{\pi_0} + \Delta_0)}\right\}.$$

*where $\Delta_0 = V^\star - V^{\pi_0}$ is the sub-optimality gap of policy $\pi_0$.*

Theorem 1 provides a general framework deriving lower-bounds for conservative exploration and highlights the impact of the baseline policy on the regret. In particular, it shows that in any sequential decision-making problem, after a sufficiently large number of episodes the conservative condition can be verified and the baseline policy has no impact anymore on the learning process. The only requirement is the knowledge of a lower-bound for the non-conservative case. Before instantiating the result in specific settings, we provide an intuition about how this result is derived and what is the role of the conservative budget $\mathcal{B}$.

**Proof Sketch.** Let us consider a conservative algorithm $\mathfrak{A} = \{\pi_t \mid t \leq T\}$, which is associated to a non-conservative algorithm $\widetilde{\mathfrak{A}} = \{\pi_t \mid t \in \mathcal{T}_0^c\}$ with $\mathcal{T}_0^c$ and $\mathcal{T}_0$ the set of non-conservative and conservative episodes as defined in Sec. 3. Now if $\mathbb{E}\,|\mathcal{T}_0| \geq \frac{\xi^2}{\alpha V^{\pi_0} \cdot (\alpha V^{\pi_0} + \Delta_0)}$ (i.e. the algorithm plays $\pi_0$ too many times), then the regret caused by $\pi_0$ is at least $\frac{\xi^2 \Delta_0}{\alpha V^{\pi_0} \cdot (\alpha V^{\pi_0} + \Delta_0)}$. When $\mathbb{E}\,|\mathcal{T}_0| < \frac{\xi^2}{\alpha V^{\pi_0} \cdot (\alpha V^{\pi_0} + \Delta_0)}$, consider the budget of $\mathcal{T}_0$ defined in Definition 1:

$$B_{\mathcal{T}_0^c}(\mathcal{A}_c) = \max_{t \in \mathcal{T}_0^c} \mathbb{E} \sum_{k=1}^{t} [(1-\alpha)V^{\pi_0} - V^{\pi^t}] = \max_{t \in \mathcal{T}_0^c} \mathbb{E}[R_{\mathfrak{A}}^{T_0^c}(\mathcal{A}_c)(t)] - (\alpha V^{\pi_0} + \Delta_0)t, \quad (6)$$

where $\mathbb{E}\left[R_{\mathfrak{A}}^{T_0^c}(\mathcal{A}_c)(t)\right]$ is the regret incurred by the rounds in $T_0^c$. Now if $\mathbb{E}\left[R_{\mathfrak{A}}^{T_0^c}(\mathcal{A}_c)(t)\right] \geq \xi\sqrt{t}$, we have $B_{\mathcal{T}_0^c}(\mathcal{A}_c) \gtrsim \frac{\xi^2}{\alpha V^{\pi_0} + \Delta_0}$ by taking maximum on the right handside of (6) (viewing RHS as a quadratic function of $\sqrt{t}$). Therefore $\mathbb{E}\,|\mathcal{T}_0| \geq \frac{B_{\mathcal{T}_0^c}(\mathcal{A}_c)}{\alpha V^{\pi_0}} \gtrsim \frac{\xi^2}{\alpha V^{\pi_0} \cdot (\alpha V^{\pi_0} + \Delta_0)}$ and the regret is also no smaller than $\frac{\xi^2 \Delta_0}{\alpha V^{\pi_0} \cdot (\alpha V^{\pi_0} + \Delta_0)}$, which completes the proof.

**Example of Lower Bounds.** For instance, in the multi-armed bandits, by leveraging the lower-bound in (Thm. 15.2 Lattimore & Szepesvári, 2020), we can obtain the following corollary of Theorem 1. This result is more general than the lower bound in Wu et al. (2016) where they have a restriction that $\frac{\Delta_0}{\alpha\mu_0 + \Delta_0} \geq 0.9$. See Appendix E for details.

**Corollary 1.** *For any $K \in \mathbb{N}^\star$, $\alpha \in [0,1]$, $\mu_0 \in [0,1]$, $\delta \in (0,1)$ and a conservative algorithm $\mathfrak{A}$ then there exists $\mu \in [0,1]^K$ such that $\sum_{l=1}^{t} \mu_{\pi_l} \geq (1-\alpha)\mu_0 t$ with high probability for any $t \leq T$. Then, for $T \geq \frac{A}{\alpha\mu_0 \cdot (\alpha\mu_0 + \Delta_0)} + \frac{\sqrt{A}}{\alpha\mu_0 + \Delta_0}$, $R_T(\mu, \mathfrak{A}) \gtrsim \max\left\{\sqrt{AT}, \frac{A\Delta_0}{\alpha\mu_0 \cdot (\alpha\mu_0 + \Delta_0)}\right\}$.*

The generality of Theorem 1 allows us to derive lower-bounds for conservative exploration in many different problems, where the lower-bound was unknown. Table 1 reports the lower-bound obtained through Theorem 1. Please refer to Appendix B for lower-bounds for non-conservative exploration. In linear bandits, the lower bound we obtain matches the result in (Kazerouni et al., 2016; Garcelon et al., 2020b), showing the optimality of their algorithms. In tabular MDPs, our result shows that the dependence on $S, A$ and $H$ of CUCBVI (Garcelon et al., 2020a) is not optimal. Finally, by instantiating Theorem 1 in low-rank MDPs, we obtain the first lower bound for this setting.

## 5 UPPER BOUNDS

In this section, we show how to leverage the framework of Sec. 3 to derive an algorithm for any conservative sequential decision-making problem. We first show that when knowing $\Delta_0$ a simple algorithm achieves a minimax regret, as prescribed by our lower bound of Sec. 4. Then, we show how to remove this knowledge without hurting the performance by combining our framework and the idea of lower confidence bound.

### 5.1 THE BUDGET-EXPORATION ALGORITHM

Given a non-conservative algorithm $\widetilde{\mathfrak{A}}$, the minimum amount of rewards needed to play this non-conservative algorithm for $T$ consecutive steps is the budget defined in Def. 1. Indeed, if we denote by $\{\tilde{\pi}_l \mid l \leq T\}$ the sequence of non-conservative policies executed by $\widetilde{\mathfrak{A}}$, then for any set $\mathcal{O} \subset [T]$ the budget can be rewritten as:

$$\mathcal{B}_T(\mathcal{O}, \{\tilde{\pi}_l \mid l \leq T\}) = \max_{t \in \mathcal{O}} \sum_{l \in \mathcal{O} \cap [t]} (1-\alpha)V^{\pi_0} - V^{\tilde{\pi}_l}$$

---

**Algorithm 1:** Budget-Exporation

---

**Input:** A non-conservative algorithm $\widetilde{\mathfrak{A}}$, conservative policy cumulative reward $V^{\pi_0}$,
conservative level: $\alpha \in (0,1)$ ,baseline action gap: $\Delta_0 = V^\star - V^{\pi_0}$ and a constant $C$

1 Set $B = \frac{C^2}{\alpha V^{\pi_0} + \Delta_0}$ and $T_0 = \frac{B}{\alpha V^{\pi_0}}$;
2 **for** $t = 1, \ldots, T$ **do**
3     **if** $t < T_0$ **then**
4        Play $\pi_0$;
5     **else**
6        Play according to $\widetilde{\mathfrak{A}}$;
7     **end**
8 **end**

---

$$= \max_{t \in \mathcal{O}} \sum_{l \in \mathcal{O} \cap [t]} \left( V^\star - V^{\pi_l} - (\Delta_0 + \alpha V^{\pi_0}) \big| \mathcal{O} \cap [t] \big| \right).$$

Let's define $R_{\mathcal{O} \cap [t]}(\widetilde{\mathfrak{A}}) := \sum_{l \in \mathcal{O} \cap [t]} V^\star - V^{\pi_l}$ the regret over the time steps in $\mathcal{O}$ of the non-conservative algorithm $\widetilde{\mathfrak{A}}$. For most non-conservative algorithms with minimax regret bound, $\tilde{R}_T(\widetilde{\mathfrak{A}}, \mathcal{O}) = \mathcal{O}(C\sqrt{|\mathcal{O} \cap [t]|})$ w.h.p., where $C \in \mathbb{R}$ is a problem-dependent quantity as in Theorem 1. For example, in multi-armed bandit $C = \sqrt{A}$ for the UCB algorithm or $C = \sqrt{H^3 S A}$ for the UCBVI-BF algorithm (Azar et al., 2017). This implies that the budget required by $\widetilde{\mathfrak{A}}$ is at least $\frac{C^2}{\Delta_0 + \alpha V^{\pi_0}}$. Therefore, the simple algorithm playing the baseline policy for the first $T_0 := O(\frac{C^2}{(\alpha V^{\pi_0} + \Delta_0)\alpha V^{\pi_0}})$ steps and then running the non-conservative algorithm $\widetilde{\mathfrak{A}}$, is conservative. We call such algorithm Budget-Exporation (see Alg. 1). This algorithm is conservative and minimax optimal. Indeed, we can show (see Theorem 2) that the regret upper bound of Budget-Exporation matches the lower bounds of Section 4. While knowing $\Delta_0$ in advance may be a restrictive assumption, it is interesting that a two-stage algorithm structure (deploying a baseline policy and then a non-conservative policy) is enough to achieve minimax optimality.

**Theorem 2.** *Consider an algorithm $\widetilde{\mathfrak{A}}$, $\delta \in (0,1)$ and constant $C \in \mathbb{R}$ such that with probability at least $1 - \delta$, for any $T \geq 1$, $R_T(\widetilde{\mathfrak{A}}) \leq \widetilde{O}(C\sqrt{T})$. Then for any $T \geq 1$, the regret of Budget-Exporation is bounded with probability at least $1 - \delta$ by $\widetilde{O}(C\sqrt{T} + \frac{C^2 \Delta_0}{\alpha V^{\pi_0}(\alpha V^{\pi_0} + \Delta_0)})$.*

Instantiating Thm. 1 with $\widetilde{\mathfrak{A}}$ being the UCB algorithm (Lattimore & Szepesvári, 2020), then $C = \sqrt{A}$ and the regret of Budget-Exporation is bounded w.h.p. by $\widetilde{O}(\sqrt{AT} + \frac{A\Delta_0}{\alpha V^{\pi_0}(\alpha \mu_0 + \Delta_0)})$, that matches our novel lower bound introduced in Sec. 4. Similar results can be obtained for the other settings, see Table 1. In linear bandit we consider LinUCB as the non-conservative algorithm, leading to $C = d$. Similarly, in tabular MDP and low-ran MDPs, we get $C = \sqrt{H^3 S A}$ and $C = \sqrt{d^3 H^4}$ respectively using UCBVI-BF (Azar et al., 2017) and LSVI-UCB (Jin et al., 2020). Refer to Table 1 for a complete comparison of the results.

## 5.2 THE LCBCE ALGORITHM

When $\Delta_0$ is unknown, we aim to use the same idea as Budget-Exporation, that is to say to play a policy different than the baseline one only if the budget is positive. To achieve this, we need to build an online estimate of the conservative budget which amounts to build a lower confidence bound (w.h.p.) on the value function of any policy $\pi$. Therefore, assuming a non-conservative algorithm $\widetilde{\mathfrak{A}}$ builds such confidence bounds, for example by estimate the MDP as done by Garcelon et al. (2020a), we show how our budget framework helps to derive a conservative regret bound.

Let's consider a non-conservative algorithm $\widetilde{\mathfrak{A}} = \{\pi_t \mid t \leq T\}$ able to construct a high probability lower bound on the set of selected policies. That is, for any time $t \leq T$ and $\delta \in (0,1)$, $\widetilde{\mathfrak{A}}$ computes a sequence of real numbers $(\lambda_t^{\pi_k}(\delta))_{k \leq t}$ such that with probability at least $1 - \delta$, for all $k \leq t$, $\lambda_t^{\pi_k}(\delta) \leq V^{\pi_k}$. Using these lower bounds, we can define a proxy to the budget for $\widetilde{\mathcal{B}}_{T,\delta}(\mathcal{O}, \widetilde{\mathfrak{A}})$ for

---

**Algorithm 2:** Lower Confidence Bound for Conservative Exploration

---

**Input:** A non-conservative algorithm $\widetilde{\mathfrak{A}}$, $\delta \in (0,1)$, lower confidence bounds $\lambda_t^{\pi_k} \leq V^{\pi_k}$, conservative policy value $V^{\pi_0}$, $\alpha \in (0,1)$

1   Set $B = 0$ ;        `// the accumulated budget`
2   Set $t' = 0$ ;     `// the number of steps in which the agent acts as` $\widetilde{\mathfrak{A}}$
3   **for** $t = 1, 2, ..., T$ **do**
4      $\widetilde{\mathfrak{A}}$ gives lower bound $\lambda_{t'+1}$ and a policy $\tilde{\pi}_{t'+1}$;
5      Set $\lambda = \sum_{k=1}^{t'} \lambda_{t'+1}^{\tilde{\pi}_k} + \lambda_{t'+1}^{\tilde{\pi}_{t'+1}}$ ;   `// lower bound of expected total reward`
6      **if** $\lambda - (t'+1)\alpha V^{\pi_0} < B$ **then**
7         Play $\pi_t = \pi_0$ and set $B = B + \alpha V^{\pi_0}$;
8      **else**
9         Play $\pi_t = \tilde{\pi}_{t'+1}$ and set $t' = t' + 1$;
10      **end**
11 **end**

---

any subset $\mathcal{O} \subset [T]$ by

$$\widetilde{B}_{T,\delta}\left(\mathcal{O}, \widetilde{\mathfrak{A}}\right) = \max_{t \in \mathcal{O}} \sum_{l \in \mathcal{O} \cap [t]} \left((1-\alpha)V^{\pi_0} - \lambda_t^{\pi_l}(\delta)\right), \tag{7}$$

with $(\pi_l)_{l \in O}$ the sequence of policies computed by the non-conservative algorithm $\widetilde{\mathfrak{A}}$. Then following from the definition of $(\lambda_t^{\pi_l}(\delta))_{l \leq t}$, we have that with probability at least $1 - \delta$ that $\widetilde{B}_{T,\delta}\left(O, \widetilde{\mathfrak{A}}\right) \geq \mathcal{B}_T(\mathcal{O}, \widetilde{\mathfrak{A}})$. This shows that it is possible to compute $\widetilde{B}_{T,\delta}\left(O, \widetilde{\mathfrak{A}}\right)$ without knowledge of the environment and the baseline parameters. The idea of our algorithm is now to play a non-conservative policy $\pi_t$ at time $t$ only if the difference between the proxy to the budget of $\widetilde{\mathfrak{A}}$ and the reward accumulated by playing the baseline policy is negative. Formally, the condition is $\widetilde{B}_{t,\delta}\left(S_t \cup t, \widetilde{\mathfrak{A}}\right) \leq \alpha V^{\pi_0}(t - 1 - |S_t|)$ where $S_t$ is the set of time step where a non-conservative policy was deployed in episodes before $t$. As a result, the minimum budget that $\widetilde{\mathfrak{A}}$ requires to be conservative is $\max_t \widetilde{B}_{t,\delta}\left(S_t \cup t, \widetilde{\mathfrak{A}}\right) = \max_{t \in [T]} \sum_{l \in S_t} \left((1-\alpha)V^{\pi_0} - \lambda_t^{\pi_l}(\delta)\right)$. The algorithm, called *Lower Confidence Bound for Conservative Exploration* (LCBCE), is detailed in Alg. 2.

Next, we show the regret bound of LCBCE. The proof is in Appendix D.

**Theorem 3.** *Consider an algorithm $\widetilde{\mathfrak{A}}$, $\delta \in (0,1)$ and constant $C \in \mathbb{R}$ such that with probability at least $1 - \delta$, for any $T \geq 1$, $R_T(\widetilde{\mathfrak{A}}) \leq \widetilde{O}(C\sqrt{T})$. If $\widetilde{\mathfrak{A}}$ computes lower confidence bound such that $\sum_{k=1}^t \left(V^{\pi_k} - \lambda_t^{\pi_k}\right) \leq \widetilde{O}(C\sqrt{T})$ with probability at least $1 - \delta$, then for any $T \geq 1$, the regret of LCBCE is bounded with probability at least $1 - \delta$ by $\widetilde{O}(C\sqrt{T} + \frac{C^2 \Delta_0}{\alpha V^{\pi_0}(\alpha V^{\pi_0} + \Delta_0)})$.*

In the MAB and tabular case, LCBCE paired with UCB achieves a better regret bound compared with previous papers(Garcelon et al., 2020a; Wu et al., 2016). We also provide the first minimax optimal bound for the case of unknown baseline parameters. Finally, in low rank MDPs we recover the same rate as in the case of known baseline. See Table 1.

## 6   CONCLUSION

We present a unified framework for conservative exploration in sequential decision-making problems. This framework can be leveraged to derive both minimax lower and upper bounds. In bandits, we provide novel lower bounds that highlighted the optimality of existing algorithms. In RL, we provide the first lower bound for tabular MDPs and a matching upper bounds, and the first analysis for low rank MDPs. An interesting question is whether one can leverage this framework to derive problem-dependent logarithmic bounds for conservative exploration. Another direction is to investigate model-free algorithms (e.g., Q-learning (Jin et al., 2018)) for conservative exploration.

ACKNOWLEDGEMENTS

Liwei Wang was supported by National Key R&D Program of China (2018YFB1402600), Exploratory Research Project of Zhejiang Lab (No. 2022RC0AN02), BJNSF (L172037). Project 2020BD006 supported by PKUBaidu Fund.

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
