# OpenReview forum: "A Reduction-Based Framework for Conservative Bandits and Reinforcement Learning"
_ICLR.cc/2022/Conference — ICLR 2022 Poster_

### Official Review · Reviewer_nbyJ · 2021-10-18

**Correctness:** 3
**Technical Novelty And Significance:** 3
**Empirical Novelty And Significance:** 1
**Recommendation:** 6
**Confidence:** 4

**Main Review:**

**strengths:**
1. The paper is clearly written and easy to follow.
2. The budget approach is simple and intuitive.
3. The regret upper bounds are optimal (near-optimal for linear MDPs but this is the best one can hope for).
4. The reduction approach provides lower bounds easily for many settings.

**weaknesses:**
1. The authors give too much credit to their MAB lower bound in my opinion. Notice that this lower bound is less tight than the previous one, and that it can be deduced from the previous lower bound and this paper's new upper bound. I think that the authors need to state this a little more accurately and focus on the lower bounds that they provide for the other settings (for which there were no previous lower bounds).
2. Although there are no previous lower bounds for conservative linear bandits/tabular MDPs/linear MDPs, I think that the authors should discuss standard constructions. That is, if I take the construction of Wu et al. and embed it in linear bandit or MDP what would I get? Embedding MAB in MDP or linear bandit is common in sequential decision making problems and should be addressed in my opinion. Could the authors explain if this is not possible here or if it gives sub-optimal lower bounds? The derivation of lower bounds through the reduction is nice but it should be put in perspective against common approaches.
3. The lower bound proof sketch is not clear at all in my opinion. I read the proof in the appendix and it was very clear so I encourage the authors to rewrite the proof sketch more closely to the way that it is written in the appendix (this is not a long proof).
4. There is no proof sketch for Theorem 3. This is the most important theorem and there is no explanation regarding its proof. The way that the LCB is combined with the budget was very interesting to me, and I was very disappointed that the (short) proof from the appendix was not sketched at all.
5. The authors mention safe RL and constrained MDP but do not explain what is the difference between them and conservative RL. These problems seem very related and I think a comparison or at least an explanation is in order.
6. The authors claim that the concept of budget was already used in Wu et al., but that it is less general because it is for the UCB algorithm. Is it really less general or simply analyzed better in the current paper (since the reduction also uses confidence bounds I am guessing it will only work with optimistic algorithms)? I am wondering whether the budget approach is novel in this paper or does this paper just analyze it better (which is also a good contribution but should be stated accurately).
7. The authors also mention budget in the context of the algorithms by Garcelon et al. and Kazerouni et al.. They say that their approach is different, but then just say that their budget is tighter. So is it really different or is the budget analyzed better in the current paper?

**Summary Of The Paper:**

This paper studies online sequential decision making under a conservative constraint, i.e., the agent needs to perform at least as well as some baseline policy. The authors present algorithms based on a reduction from non-conservative MAB/linear bandits/tabular MDPs/linear MDPs, and prove minimax optimal (or near-optimal) upper and lower bounds on the regret.

**Summary Of The Review:**

The paper studies an interesting question and presents a natural reduction-based approach that yields optimal upper and lower bounds. However, the relation to previous work is not described clearly, the proof sketches are either unclear or missing and the authors put their focus on a lower bound that is less tight than previous ones. Overall, while this paper has interesting contributions, it requires further revision before being published in my opinion.

---

> ### Author Response · Authors · 2021-11-17
> **Response to Reviewer nbyJ**
>
> We thank the reviewer for your valuable comments. We **address all of your concerns** below:
>
> 1. **The lower bound is not as tight as the one in [Wu et al. 2016]**: In the revised version, we gave a thorough technical comparison with [Wu et al. 2016]’s lower bound in Section E. In summary  (we focus on the term that related to the conservative constraint):
>
> - The lower bound in Wu et al. 2016 can be stated as: for any given $V^{\pi_0}$ and $\alpha$, **there exists** a $\Delta_0$ that satisfies $0.9 < \frac{\Delta_0}{\alpha V^{\pi_0}+\Delta_0} < 1$ such that the regret is lower bounded by $\Omega(\frac{A}{\alpha \mu_0})$.
>
> - Our lower bound can be stated as: for **any given** $V^{\pi_0}$, $\alpha$, and $\Delta_0$, the regret is lower bounded by $\Omega(\frac{A\Delta_0}{\alpha \mu_0(\alpha \mu_0 + \Delta_0)})$.
> Note if the given $\Delta_0$ satisfies $0.9 < \frac{\Delta_0}{\alpha V^{\pi_0}+\Delta_0} < 1$, then our bound recovers theirs. However, our bound does not require this condition and thus is more general.
>
> 2. **Embedding MAB lower bounds to linear bandits / RL**: Simply embedding MAB lower bounds to linear bandits / RL will give a suboptimal lower bound.  For example, if we embed a conservative MAB problem with $d$ arms in a $|d|$-dimensional linear bandit problem, we will only get a sub-optimal lower bound $\Omega(\sqrt{dT}+d/(\alpha\mu_{0}))$. We note that generally, embedding a construction for MAB to linear bandits / tabular MDP will only give suboptimal lower bounds **even in the non-conservative setting**. See, e.g., Chapter 24 of Lattimore & Szepesvari (2020) and Domingues et al. (2021). We have also added some discussions in Section 1.2.1 in the revised version.
>
> 3.  **Lower bound proof sketch**: Thanks for your suggestion. We have rewritten our proof sketch for Theorem 1. We hope it will help the readers understand the theorem more easily.
>
> 4. **Proof for Theorem 3**: Thanks for your advice. We have moved our proof for Theorem 3 to the main text.
>
> 5. **Safe bandits/RL, constrained bandits/RL**: we have added more discussions in the revised version. See the last paragraph of Section 1.3.
>
> 6. **Comparisons with Wu et al., Garcelon et al, and Kazerouni et al.**: Yes, you are right that our improvement is mainly due to a better analysis. We have rewritten Section 1.2 to better describe our technical innovations.
>
> - For the **known $\Delta_0$ case**, when specializing our Budget-Exploration algorithm to the MAB setting, we recover the BudgetFirst algorithm in Wu et al. (2016). However, our regret bound is tighter. Their analysis on the required budget is loose: they accumulate enough budget to play **$T$-step** exploration where $T$ is the total number of rounds. Our main technical insight to obtain the tight regret bound is a sharp analysis of the required budget: we **relate the minimax regret upper bounds of UCB algorithms and show the required budget can be independent of $T$.** Also see Section F for more technical details.
>
>
> - For the **unknown $\Delta_0$ case**, when our LCBCE  algorithm is specialized to the MAB setting, we recover Wu et al.’s algorithm, when specialized to linear bandits, we recover Kazerouni et al’s algorithm, and when specialized to tabular MDP, we recover Garcelon et al. (2020a)’s algorithm. We note that all these works adopt the same algorithmic template: 1) build an online estimate on the lower bound performance of each possible exploration policy, and 2) based on the estimated lower bounds, choose an exploration policy or play the baseline policy.  The key difference and the most non-trivial part in different papers is how to analyze $T_0$ (the number of times of executing the baseline policy). Existing works upper bound $T_0$ by relating it to the decision criterion for whether to choose the baseline policy or not. Since for different problem settings, the criteria have different forms, existing papers adopt different problem-specific analyses, and in some settings, the analyses are not tight (e.g., MAB and tabular RL).
> Our analysis approach is different from existing ones: we bound $T_0$ via **maximizing a quadratic function that depends on the minimax regret bounds of non-conservative algorithms and the conservative constraint**.
>
> We hope our response and revision clarify your concerns and you will consider raising your score. Thank you!

---

> > ### Comment · Reviewer_nbyJ · 2021-11-29
> > **Raising my score**
> >
> > I thank the reviewers for the clarifications and raising my score to 6.

---

### Official Review · Reviewer_g4nC · 2021-10-28

**Correctness:** 4
**Technical Novelty And Significance:** 3
**Empirical Novelty And Significance:** Not applicable
**Recommendation:** 8
**Confidence:** 4

**Main Review:**

This paper is well written. The studied problem is significant. The generality of the results are impressive.

While the reviewer tends to accept the paper, the reviewers has some concerns.

1/ No assumption is made on the baseline \pi_0. However, Budget-Exploration plays \pi_0 during a first period of time. What does happen if \pi_0 plays only one sub-optimal arm? How can learn the non-conservative baseline ?

2/ The proposed lower bound is not as tight the one in (Wu et al 2016): the right term is greater in (Wu et al 2016). However, Theorem 3 states that the regret upper bound of LCBCE reaches the proposed lower bound. Hence it violates the lower stated in (Wu et al 2016). Could you comment? (The reviewer does not understand the comment below Corollary 1).

3/ The reviewer thinks that the comment on the comparison with (Wu et al 2016) above section 1.3 is not supported by the facts. In particular the reviewer does not get the sentence “our algorithm calculates how much budget the algorithm needs in advance, and therefore has a tighter regret bound”. First it only concerns Budget-Exploration. Second generally an algorithm that uses the seen data is better than an algorithm that does not use them.


AFTER REBUTTAL
The authors have answered my concerns. I vote for acceptance.

**Summary Of The Paper:**

This paper proposes a reduction-based framework for a large class of reinforcement learning algorithms, including bandits, linear bandits, tabular MDP and linear MDP.
The authors notably propose a generic lower bound that holds for all the studied class of algorithms. The lower bound is built on the regret decomposition between the regret of the conservative baseline during the time when the budget is not reached, and the regret of a non-conservative algorithm that learns on the baseline. While the obtained lower in the bandit setting is not as tight as the one obtained in (Wu et al, 2016), this lower bound holds for a larger class of algorithms.
Then, two generic algorithms are proposed for handling conservative reinforcement learning. Budget-Exploration consists in learning the non-conservative algorithm on the conservative baseline during a fixed period of time and then to play the non-conservative algorithm. The regret upper bound of Budget-Exploration matches the lower bound, but the knowledge of the baseline gap is need.
The second algorithm LCBCE does not necessitate that the baseline gap be known. The idea of the algorithm is to compute an upper bound of the budget thanks to the lower bounds of rewards obtained by the non-conservative algorithm run on the baseline. The regret upper bound of LCBCE still matches the proposed lower bound.


**Summary Of The Review:**

While the reviewer tends to accept the paper, the reviewers has some concerns.

---

> ### Author Response · Authors · 2021-11-17
> **Response to Reviewer g4nC**
>
> Thanks for your careful reading and the positive review.
>
> **No assumption is made on the baseline $\pi_0$**: We note that it actually doesn’t matter what the conservative policy $\pi_0$ is, because the algorithm does not rely on $\pi_0$ to explore the unknown environment. All we need is to run $\pi_0$ to accumulate a budget so that the agent can use non-conservative policy to explore the environment without violating the conservative constraint.
>
> **The lower bound is not as tight as the on in [Wu et al. 2016]**: In the revised version, we gave a thorough technical comparison with [Wu et al. 2016]’s lower bound in Section E. In summary  (we focus on the second term that related to the conservative constraint):
>
> - the lower bound in Wu et al. 2016 can be stated as: for any given $V^{\pi_0}$ and $\alpha$, **there exists** a $\Delta_0$ that satisfies $0.9 < \frac{\Delta_0}{\alpha V^{\pi_0}+\Delta_0} < 1$ such that the regret is lower bounded by $\Omega(\frac{A}{\alpha \mu_0})$.
> - Our lower bound can be stated as for **any given** $V^{\pi_0}$, $\alpha$, and $\Delta_0$, the regret is lower bounded by $\Omega(\frac{A\Delta_0}{\alpha \mu_0(\alpha \mu_0 + \Delta_0)})$.
>
> Note if the given $\Delta_0$ satisfies $0.9 < \frac{\Delta_0}{\alpha V^{\pi_0}+\Delta_0} < 1$, then our bound recovers theirs. However, our bound does not require this condition and thus is more general.
>
> **Comparison with [Wu et al 2016]’s upper bounds**: we have rewritten Section 1.2 with more discussions and we added Section F on technical details. We improve Wu et al. (2016)’s regret bound with known $\Delta_0$ using a novel, tighter analysis on the required budget. Their analysis on the required budget is loose: they accumulate enough budget to play **$T$-step** exploration where $T$ is the total number of rounds. Our main technical insight to obtain the tight regret bound is a sharp analysis of the required budget: we **relate the minimax regret upper bounds of UCB algorithms and show the required budget can be independent of $T$**.
>
> We agree that, for many machine learning problems, using the seen data is better is true, but in our problem, this is not the case. The reason is that we already have a good understanding of the non-conservative algorithms (such as UCB) theoretically, therefore we can perform a tight analysis of the budget it needs in theory, and do not need to wait for seen data.

---

### Official Review · Reviewer_wDiu · 2021-11-01

**Correctness:** 4
**Technical Novelty And Significance:** 2
**Empirical Novelty And Significance:** Not applicable
**Recommendation:** 6
**Confidence:** 4

**Main Review:**

The paper is well-written in general and the contributions and the model are well-motivated. Numerical experiments that corroborate the theoretical guarantees of the paper are missing in the current version and I highly recommend to add some in the revision since the theoretical guarantees and algorithms seem to be incremental compared to the existing works.

Could the authors provide a convincing argument about the non-incremental nature of the novelty.

**Summary Of The Paper:**

This paper studies bandits and RL settings subject to a conservative constraint where the agent has to perform at least as well as a given baseline policy. It improves the existing lower bound for conservative MAB, and as the main contribution, obtains new lower bounds for conservative linear bandits, tabular RL and low-rank MDP. It also provides new upper bounds matching existing ones with different analyses.

**Summary Of The Review:**

The problem studied in this paper is interesting and I think it would interest the ICLR community. While the theoretical derivations heavily reliy on Kazerouni et al 2017 and Wu et al. 2016, the theoretical results are interesting. I have not checked all the proofs, but they seem sound and correct.

---

> ### Author Response · Authors · 2021-11-17
> **Response to Reviewer wDiu**
>
> Thanks for your careful reading and the positive review.
> We respectfully disagree with your comment
>
> > the theoretical guarantees and algorithms seem to be incremental compared to the existing works
>
> Below we give a summary of technical innovations. See Section 1.2 and Section F in the revised paper for more details. The main idea of our paper is novel and different from previous literature, such as [Wu et al., 2016], in two perspectives.
>
> **Lower Bounds**: For lower bounds, our technique is **completely different** from [Wu et al. 2016] (which is the only paper that has lower bounds). They followed a classical approach in bandit literature. They constructed a class of hard environments and used an information-theoretic argument to prove the lower bound. On the other hand, our technique is **non-constructive**. We develop a reduction method that turns a lower bound in the non-conservative setting to a lower bound in the conservative setting. Our technique yields a more general lower bound for multi-armed bandits and lower bounds for other problems.
>
> **Upper Bounds**:
>
> **Known $\Delta_0$**: we improve Wu et al. 2016’s regret bound with known $\Delta_0$ using a novel, tighter analysis on the required budget. Their analysis on the required budget is loose: they accumulate enough budget to play **$T$-step** exploration where $T$ is the total number of episodes. Our main technical insight to obtain the tight regret bound is a sharp analysis of the required budget: we **relate the minimax regret upper bounds of UCB algorithms, and show the required budget  can be independent of $T$**
>
> **Unknown $\Delta_0$**: Existing work [Wu et al., 2016, Kazerouni et al. 2016, Garcelon et al. 2020ab] and our work all adopt the same algorithmic template: 1)  build an online estimate on the lower bound performance of each possible exploration policy, and 2) based on the estimated lower bounds, choose an exploration policy or play the baseline policy.
>
> The key difference and the most non-trivial part in different papers is how to analyze $T_0$ (the number of times of executing the baseline policy). Existing works upper bound $T_0$ by relating it to the decision criterion for whether to choose the baseline policy or not. Since for different problem settings, the criteria have different forms, existing papers adopt different problem-specific analyses, and in some settings, the analyses are not tight (e.g., MAB and tabular RL).
>
> Our analysis approach is different from existing ones: we bound $T_0$ via **maximizing a quadratic function that depends on the minimax regret bounds of non-conservative algorithms and the conservative constraint**.
>
> We hope our response addresses your concern that our work is incremental.

---

### Official Review · Reviewer_yivy · 2021-11-07

**Correctness:** 4
**Technical Novelty And Significance:** 3
**Empirical Novelty And Significance:** 3
**Recommendation:** 8
**Confidence:** 3

**Main Review:**

Strengths
* The proposed algorithm LCBCE is a general framework which could be used to tackle many bandit and RL problems
* For some of the problems (multi-armed bandits, linear bandits and tabular MDPs), a nearly optimal upper bound is obtained.
* The paper is written well and the ideas are easy to understand.

Weaknesses
* Although the proposed framework seems very general, the main ideas of the paper come from the existing literatures. In the beginning, the authors propose a simple policy called Budget-Exploration which performs the baseline policy first to ensure that there is enough budget and then conducts the non-conservative policy for the remaining episodes of the game assuming that the gap between the optimal policy and the baseline policy ($\Delta_0$) is known. A similar idea is already proposed in the BudgetFirst policy in Wu et al., 2016. Later, the authors extend the idea to the case when $\Delta_0$ is unknown where they build an online estimate on the lower bound performance of the policy, which also shares a similar idea with Conservative UCB proposed in Wu et al., 2016. I find it difficult to believe the impact is significant except that the authors deploy the idea under a general problem setting.
* No experiments are conducted to illustrate the empirical performance of the proposed algorithm.

**Summary Of The Paper:**

This paper investigates bandits and reinforcement learning problems under the conservative setting where it is required that with high probability, the performance of the proposed policy is comparable to that of a baseline policy $\pi_0$. An efficient framework LCBCE together with a lower bound are proposed which are able to be used to tackle the multi-armed bandit, linear bandit, tabular MDP and even low-rank MDP problems.

**Summary Of The Review:**

Although the authors propose a general framework to deal with multi-armed bandits and RL problems, it seems to me that the work is an extension of the existing work i.e., Wu et al., 2016 to a more general problem setting.  If I read the paper correctly, I did not see any new techniques developed to tackle this more general problem setting.

Detailed comments:

__Before section 2__, ... All these works focused on provide an upper-bound to the regret of a conservative algorithm...: provide -> providing

__Notations__, ...there exists a constant $c$ such that $A \geq (\leq) cB$... : constant -> positive constant

__After Assumption 1__, ... there exists $\theta^{\pi} \in \mathbb{R}^d$ such that $Q_h^{\pi}(s, a) = \langle \phi(s, a), \theta^{\pi}_h \rangle. $: $\theta^{\pi}$ -> $\theta^{\pi}_h$

---

> ### Author Response · Authors · 2021-11-17
> **Response to Reviewer yivy**
>
> Thanks for your careful reading and the positive review.
> We respectfully disagree with your comment:
>
> > If I read the paper correctly, I did not see any new techniques developed to tackle this more general problem setting.
>
> First of all, we not only obtain sharp regret bounds for new settings but also **improve existing results of Wu et al. 2016 and  Garcelon et al. 2020a**. Improvement is generally not possible without new ideas. Below we give a summary of technical innovations. See Section 1.2 and Section F in the revised paper for more details. The main idea of our paper is novel and different from previous literature, such as [Wu et al., 2016], in two perspectives.
>
> **Lower Bounds**: For lower bounds, our technique is **completely different** from [Wu et al. 2016] (which is the only paper that has lower bounds). They followed a classical approach in bandit literature. They constructed a class of hard environments and used an information-theoretic argument to prove the lower bound. On the other hand, our technique is **non-constructive**. We develop a reduction method that turns a lower bound in the non-conservative setting to a lower bound in the conservative setting. Our technique yields a more general lower bound for multi-armed bandits and lower bounds for other problems.
>
> **Upper Bounds**
>
> **Known $\Delta_0$**: we improve Wu et al. 2016’s regret bound with known $\Delta_0$ using a novel, tighter analysis on the required budget. Their analysis on the required budget is loose: they accumulate enough budget to play **$T$-step** exploration where $T$ is the total number of rounds. Our main technical insight to obtain the tight regret bound is a sharp analysis of the required budget: we **relate the minimax regret upper bounds of UCB algorithms and show the required budget can be independent of $T$**.
>
> **Unknown $\Delta_0$**: Existing work [Wu et al., 2016, Kazerouni et al. 2016, Garcelon et al. 2020ab] and our work all adopt the same algorithmic template: 1)  build an online estimate on the lower bound performance of each possible exploration policy, and 2) based on the estimated lower bounds, choose an exploration policy or play the baseline policy.
>
> The key difference and the most non-trivial part in different papers is how to analyze $T_0$ (the number of times of executing the baseline policy). Existing works upper bound $T_0$ by relating it to the decision criterion for whether to choose the baseline policy or not. Since for different problem settings, the criteria have different forms, existing papers adopt different problem-specific analyses, and in some settings, the analyses are not tight (e.g., MAB and tabular RL).
>
> Our analysis approach is different from existing ones: we bound $T_0$ via **maximizing a quadratic function that depends on the minimax regret bounds of non-conservative algorithms and the conservative constraint**.
>
> We hope our response addresses your concern that our work has no new technique.

---

### Author Response · Authors · 2021-11-17
**Summary of Revisions**

All major revisions are marked in red. We summarize our major revisions below:
1. We rewrote Section 1.2 to make our technical innovations more explicit and add more details on our improvements over existing works. We also added a new section (Section F) on why we can improve Wu et al. (2016)’s bound with more technical details. These revisions address the comments from Reviewer ykvy and Reviewer wDiu that our paper is incremental compared with existing work and the comment from Reviewer nbyJ on the differences from existing works.
2. We added more details about safe bandits/RL and constrained bandits/RL in Section 1.3, as requested by Reviewer nbyJ.
3. We rewrote the proof sketch of Theorem 1 (lower bound) as requested by Reviewer nbyJ.
4. We added the proof of Theorem 3 (upper bound) to the main text as requested by Reviewer nbyJ.
5. We added a new section (Section E) in the appendix to give details on why **our lower bound implies the one in Wu et al. (2016), and is strictly more general**. The new section addresses a common confusion from both Reviewer g4nC and Reviewer nbyJ that our lower bound is less tight than the one in Wu et al. (2016).

---

### Author Response · Authors · 2021-11-17
**Our lower bound is strictly more general and never less tight than Wu et al. (2016)’s lower bound**

A common confusion from Reviewer g4nC and Reviewer nbyJ is that our lower bound for multi-armed bandits is less tight than the one in Wu et al. 2016.

This is not true.  In the initial version's footnote 2 and the comments below Corollary 1, we briefly discussed why their bound seems tighter.  In the revised version, we gave a thorough technical explanation in Section E.
In summary  (we focus on the term that related to the conservative constraint):
- the lower bound in Wu et al. 2016 can be stated as: for any given $V^{\pi_0}$ and $\alpha$, **there exists** a $\Delta_0$ that satisfies $0.9 < \frac{\Delta_0}{\alpha V^{\pi_0}+\Delta_0} < 1$ such that the regret is lower bounded by $\Omega(\frac{A}{\alpha \mu_0})$.
- Our lower bound can be stated as for **any given** $V^{\pi_0}$, $\alpha$, and $\Delta_0$, the regret is lower bounded by $\Omega(\frac{A\Delta_0}{\alpha \mu_0(\alpha \mu_0 + \Delta_0)})$.
Note if the given $\Delta_0$ satisfies $0.9 < \frac{\Delta_0}{\alpha V^{\pi_0}+\Delta_0} < 1$, then our bound recovers theirs. However, our bound does not require this condition and thus is more general.

Another way to see that our lower bound is more general is that **our upper bound seems tighter than their lower bound**. Note there is no contradiction because their lower bound only holds in the regime  $0.9 < \frac{\Delta_0}{\alpha V^{\pi_0}+\Delta_0} < 1$, in which case, our upper bound matches their lower bound.

---

### Decision · Program_Chairs · 2022-01-20

**Decision:**

Accept (Poster)

**Comment:**

Summary: The paper studies RL and bandits in the conservative setting where the performance of the new, learnt policy should never be significantly worse than that of a baseline.

Discussions: The main concern of the reviewers was about novelty, and specifically what new techniques and ideas were brought in this work compared to (Wu et al. 2016) and (Garcelon et al 2020). The authors have addressed these concerns and updated their draft accordingly. The reviewers have now all reached a consensus and recommend to accept this work.

Recommendation: Accept